# Analysis of Volatile Compounds in Pears by HS-SPME-GC×GC-TOFMS

**DOI:** 10.3390/molecules24091795

**Published:** 2019-05-09

**Authors:** Chenchen Wang, Wenjun Zhang, Huidong Li, Jiangsheng Mao, Changying Guo, Ruiyan Ding, Ying Wang, Liping Fang, Zilei Chen, Guosheng Yang

**Affiliations:** 1Institution of Quality Standard and Testing Technology for Agro-Product, Shandong Academy of Agricultural Science, Jinan 250100, China; wangchenchen0826@163.com (C.W.); zipingguozhang@163.com (W.Z.); lihuidong8066@163.com (H.L.); maojiangsheng@163.com (J.M.); cyguo808@163.com (C.G.); zengding-1978@163.com (R.D.); lpfang922@163.com (L.F.); 2Shandong Provincial Key Laboratory of Testing Technology for Food Quality and Security, Jinan 250100, China; 3Department of Bioengineering, Qilu University of Technology, Jinan 250353, China; hanyan1226@126.com; 4College of Life Science, Shandong Normal University, Jinan 250014, China; 5School of Chemistry and Chemical Engineering, Shandong University, Jinan 250100, China

**Keywords:** pears, HS-SPME, volatile compounds, GC×GC-TOFMS

## Abstract

Aroma plays an important role in fruit quality and varies among different fruit cultivars. In this study, a sensitive and accurate method based on headspace solid-phase microextraction (HS-SPME) coupled with comprehensive two-dimensional gas chromatography time-of-flight mass spectrometry (GC×GC-TOFMS) was developed to comprehensively compare aroma components of five pear cultivars. In total, 241 volatile compounds were identified and the predominant volatile compounds were esters (101 compounds), followed by alcohols (20 compounds) and aldehydes (28 compounds). The longyuanyangli has the highest relative concentration (838.12 ng/g), while the Packham has the lowest (208.45 ng/g). This study provides a practical method for pear aroma analysis using SPME and GC×GC-TOFMS.

## 1. Introduction

Pear (*Pyrus spp.*, Rosaceae) is a popular fruit and is extensively grown in China. In 2017, based on the FAO Statistical Database, 16,527,694 tonnes of pears were produced in China taking up 68.38% of the total pear production in the world [1]. The pear cultivars mainly belong to 4 types, *P. communis* L., *P. pyrifolia* (Burm.) Nakai, *P. ussuriensis* Max. and *P. bretschneideri* Redh [2]. The *P. sinkiangensis* Yu. was reported as the fifth pear category.

Fruit aroma is one of the most important factors contributing to the overall flavor and consumer preference [2]. Therefore, several studies have investigated the aroma components of different pear cultivars [2,3,4,5,6,7]. The aroma compounds of pears are complicated and vary among pear cultivars. Most Occidental pears have intense aromas and juicy texture, whereas *P. bretschneideri* cultivars are characterized by their faint odor and crisp texture [2]. Investigation of pear aromas has focused on composition changes among pear cultivars [3,4,5,6,7], storage conditions influences [8,9,10,11,12] and postharvest treatment [11,13]. Low temperature conditioning [12], calcium treatment [13], ultralow oxygen environment [10,11] and 1-methylcyclopropene (1-MCP) treatment [11] are external factors that affect pear aroma formation and emission. Volatile compounds of pears include esters, aldehydes, alcohols, ketones and hydrocarbons. Esters are the major volatile components in *P. ussuriensis* and *P. communis*, while in *P. pyrifolia* aldehydes are the dominant volatile compounds, followed by alcohols and esters [3,4,5]. C_6_ compounds (C_6_ aldehydes and C_6_ alcohols) that were reported to be significant components in fruits [14,15,16] were also detected in pears [3,4].

Volatiles emitted from pear fruits have been studied by SPME and gas chromatography-mass spectrometry (GC-MS) in recent years. SPME makes great contributions to volatile compounds analysis [17]. However, investigations using GC-MS only identified a small quantity of volatile compounds. In comparison with one-dimensional gas chromatography (1D-GC), comprehensive two-dimensional gas chromatography (GC×GC) can provide significant signal enhancement and a 5-fold to 15-fold improvement in peak capacity [18,19]. GC×GC-TOFMS has been applied to volatiles identification such as wines [20], cloud waters [21] and green teas [22], but pears are not included.

In this research, HS-SPME and GC×GC-TOFMS were used to analyse volatile compounds in five pear cultivars. Packham’s Triumph, Docteur Jules Guyot, Clapp’s Favorite and Starkrimson are four occidental pears, which are introduced from abroad. Longyuanyangli is a hybrid variety that has intense aroma. Therefore in this study, the aromas of five pear cultivars are comprehensively investigated.

## 2. Results and Discussion

### 2.1. Optimization of the Modulation Period

Modulation period is a parameter of crucial importance in GC×GC-TOFMS analysis. Modulator is responsible to trap and refocus the components from the 1D-column and transfer them to the 2D column for further separation [18]. The modulator makes the GC×GC possible. For aroma components analysis, the modulation period was set as 2 s, 3 s, and 4 s. The shorter the modulator period is, the narrower chromatography band and the higher peak capacity are obtained. As shown in Figure 1, when the modulation period was 3 s, the retention time of 2-methylnaphthalene and 1-methylnaphthalene was 1185 s, 2.00 s and 1212 s, 2.08 s, respectively. When the modulation period was 2 s, the retention time of 2-methylnaphthalene and 1-methylnaphthalene was 1186 s, 0.00 s and 1214 s, 0.08 s, respectively. The chromatographic peak of 2-methylnaphthalene was divided into two parts, which has a significant impact on the quantification process. The peak of 1-methylnaphthalene was at the very bottom in the chromatography. In order to ensure the peak shape of the volatile compounds and the accuracy of the quantitative analysis, the modulation period was set for 3 s with a 0.6 s hot pulse time.

### 2.2. SPME Fibre Selection

Pear aromas are extremely complex and may comprise hundreds of constitutes of different physical and chemical properties. Therefore, four SPME fibres coated with different stationary phases were compared for extraction of volatile compounds. They are 100 μm PDMS (nonpolar), 85 μm PA (polar), 65 μm PDMS/PVB (bipolar) and 50/30 μm DVB/CAR/PDMS (bipolar). The aroma components of a same yali pear (*P. bertschneideri* Reld) were analyzed for fibres comparison. The aroma extraction process was repeated for three times to guarantee the accuracy of the results. Figure 2 illustrates the peak numbers and the average peak areas for volatile compounds extracted from yali pear using different SPME fibres. A total of 146 and 163 volatile compounds were identified using 65 μm PDMS/PVB and 50/30 μm DVB/CAR/PDMS, respectively. The fewest compounds were extracted by 85 μm PA fibre. In contrast to PDMS/PVB fibre, the use of DVB/CAR/PDMS fibre can obtain higher peak areas. The peak numbers of different classes obtained using four SPME fibres were shown in Appendix A. These results indicated that the 50/30 μm DVB/CAR/PDMS fibre was the optimum for extracting volatile compounds from pears. Therefore, 50/30 μm DVB/CAR/PDMS fibre was selected for aroma extraction in this study.

### 2.3. Volatile Compounds

The 2D chromatography of five pears obtained after HS-SPME-GC×GC-TOFMS analysis is shown in Appendix A. The colour gradient reflects the intensity of the TOFMS signal from low (blue) to high (red). In this study, 241 volatile compounds were tentatively identified, including 101 esters, 30 alkenes, 12 alkanes, 19 arenes, 28 aldehydes, 8 ketones, 20 alcohols and 23 others compounds. The volatile compounds amounts show great variation in different pear cultivars and ranged from 67 compounds in Packham to 160 compounds in longyuanyangli. The number of chemical classes of each pear is shown in Figure 3. Figure 4 shows that the percentage contents of volatile compounds in pears are of large differences. Esters are the dominant aromas in pears, followed by alcohols and aldehydes. Appendix A summarizes the volatile compounds detected in five pear cultivars. In this study, the retention time of *n*-alkanes (C_5_–C_20_) was obtained and the retention index of each volatile compound was calculated. The ChromaTOF-GC uses the Van den Dool and Kratz equation for the calculation of the retention index. The equation is:
$$RI_{a}=\left(\frac{RT_{a}-RT_{n}}{RT_{N}-RT_{n}}\right)100\left(N-n\right)+100RT_{n}$$
*RI_a_*: the retention index of the compound of interest; *a*: the compound of interest; *n*: the carbon number of the lower normal alkane; *N*: the carbon number of the higher normal alkane; *RT*: the retention time.

#### 2.3.1. Esters

Esters are the dominant compounds in pears. A total of 76 esters were identified in longyuanyangli. Followed by Stark (55), Clapp (51), D Jules (45), and Packham, who had the fewest esters (23) (Figure 3). The longyuanyangli has the highest concentration of esters (692.72 ng/g, 82.65%), while the Packham has the lowest (162.66 ng/g, 78.03%) (Appendix A). Acetates with high concentrations were the major ester constituents, including methyl acetate, ethyl Acetate, n-propyl acetate, butyl acetate, pentyl acetate, hexyl acetate and heptyl acetate (Appendix A). Sulfur-containing compounds, ethyl 3-(methylsulfanyl)propanoate, 3-(methylthio)propyl acetate and ethyl 3-(methylthio)-(*E*)-2-propenoate were also detected in this study. Sulfur-containing compounds have been reported to have originated from methionine and cysteine [3] and provided the juicy, fresh aroma to many fruits [23]. Methyl (*E*,*Z*)-2,4-decadienoate and ethyl (*E*,*Z*)-2,4-decadienoate are two esters that have a pear-like smell and are major volatile compounds existed in Bartlett [10] and Beurre Bosc [3]. In this study, methyl (*E*,*Z*)-2,4-decadienoate and ethyl (*E*,*Z*)-2,4-decadienoate were also detected in longyuanyangli and Packham, but the contents (less than 0.53 ng/g) were very low. Furthermore, long-chain aliphatic acid esters such as methyl tetradecanoate, methyl hexadecanoate, methyl (*Z*)-9-octadecenoate were also detected in this study.

#### 2.3.2. Alcohols

Alcohols were the second dominant volatile compounds in the five pear cultivars (Figure 4). Alcohols account for 5.33 percent (44.67 ng/g) in longyuanyangli to 13.48 percent (51.19 ng/g) in Clapp. Ethanol was the primary alcohol compounds with the concentrations ranging from 12.34 ng/g in Packham to 27.35 ng/g in D Jules. 1-Hexanol and trans-2-hexen-1-ol were reported in many fruits and regarded as C_6_ alcohols. 1-Hexanol was found in the range of 9.84 ng/g and 26.97 ng/g, but tran*s*-2-hexen-1-ol was only detected in Packham in the concentration of 0.50 ng/g (Appendix A). In addition, 1-butanol and 1-heptanol are straight-chain alcohols, which existed in all pears. Linalool, which possesses a floral and citrus-like aroma [22,24], was identified in D Jules cultivar for the first time. Citronellol was identified in D Jules, Clapp and Stark with low concentration. 

#### 2.3.3. Aldehydes and Ketones

The largest number of aldehydes compounds (24) was detected in Clapp, and Packham contained the fewest aldehydes (13) (Figure 3). The concentrations of aldehydic compounds were relatively low in all pears other than acetaldehyde, hexanal and (*E*)-2-hexenal. Acetaldehyde was detected in high concentrations ranged from 4.81 ng/g in D Jules to 23.71 ng/g in longyuanyangli. C_6_ compounds (C_6_ aldehydes and C_6_ alcohols) are regarded as green leaf volatiles and contribute to the herbaceous odour in fruits [14,15,16]. In this study, hexanal, (*E*)-2-hexenal and 1-hexanol are dominant C_6_ compounds. The concentration of hexanal ranged from 2.07 ng/g in D Jules to 12.46 ng/g in longyuanyangli and the (*E*)-2-hexenal concentration changed from 1.60 ng/g in D Jules to 6.27 ng/g in longyuanyangli. In addition, benzaldehyde was detected in all pears. Benzaldehyde has an almond-like smell and has been previously isolated from green teas [22], lychee [24], and apricot [16]. Figure 4 shows that small proportion ketones were detected. In total, eight ketones were found, but 6-methyl-5-heptene-2-one was the only ketone that presented in all pears. It has been reported that 6-methyl-5-heptene-2-one was present in higher amount in the peel compared to the flesh and was a degradation product of lycopene [16] or α-farnesene [25,26]. In addition, 6-methyl-5-heptene-2-one possesses fatty, green, citrus odour [27] and is a common ketone existed in many fruits [16,24,28,29].

#### 2.3.4. Hydrocarbons

Although 12 alkanes were identified, the relative contents were very low. A series of *n*-alkanes (C_13_–C_16_) existed in all five pears. Alkenes account for 0.56–5.88% in total volatile compounds (Figure 4). A total of 30 alkenes were detected, comprising aliphatic alkenes (5), aromatic alkenes (6) and terpenes (19). Styrene was previously identified in Chinese white pear [5]. In addition to styrene, aromatic alkenes identified in this study comprise 1-propenylbenzene, 1-ethenyl-3-ethylbenzene, 1-ethenyl-4-ethylbenzene, 1,4-dethenyl benzene, 1,3-diethenylbenzene. Terpenes which play important role in fruit flavors have been identified in pears even if their contents were much lower than other compounds. Among these terpenes, β-myrcene (Grassy, piney), (*Z*)-β-ocimene (floral, citrusy) and limonene (citrusy) are three monoterpenes [30]. Furthermore, β-myrcene has been previously identified in mango [31], apricot [16] and lychee [24]. In this study, four isomers of farnesene were detected for the first time, including (*E*)-β-farnesene, (*Z*,*E*)-α-farnesene, α-farnesene and (*Z*,*Z*)-α-farnesene. α-farnesene is the only alkenes found in five pear cultivars and accounted for the highest proportion in alkenes. (*E*)-β-farnesene, (*Z*,*E*)-α-farnesene and (*Z*,*Z*)-α-farnesene were also identified in four pear cultivars other than Packham. (*E*)-γ-bisabolene, (*Z*)-γ-bisabolene and α-humulene are major volatile compounds in carrots [32,33]. It is the first time that (*E*)-γ-bisabolene and (*Z*)-γ-bisabolene were found in pears. The two isomers existed in all pears apart from Packham. The α-humulene contributing to the woody smell [32] was identified in Packham and Clapp. Other terpenes such as α-cubenene, copaene, α-muurolene, (+)-δ-cadinene, *cis*-calamenene, α-calacorene also play important role in pear aroma.

In contrast to other chemical classes of volatiles, arenes are minor components of total volatiles. A total of 19 arenes were identified in this study. In addition to benzene and benzene homologous compounds, polycyclic aromatic hydrocarbon 2-methylnaphthalene, 1-methylnaphthalene, naphthalene were also found (Appendix A).

#### 2.3.5. Others

Pears have high concentrations of esters, alcohols, aldehydes and alkenes which are of great significance on fruit aroma. Nevertheless, other compounds (23 compounds) such as benzonitrile, 2-pentylfuran, estragole and sesquirosefuran also contribute to the overall flavor of pears and account for 0.26–1.53% of the total volatiles. Among these volatiles, three volatile acids were identified, including acetic acid, thioacetic acid and (*E*)-3-octenoic acid. Eucalyptol, which is characterized by a fresh, camphoraceous, cool odour [34] was detected in longyuanyangli in the concentration of 0.89 ng/g. Sesquirosefuran, a natural constituent existed in essential oils [35,36], was detected in D Jules, Clapp and Stark. Previous study has shown the existence of estragole in *Pyrus ussuriensis* cultivars [4], but its isomers anethole and (*Z*)-1-methoxy-4-(prop-1-en-1-yl)benzene were also identified in this study. Additionally, 2-pentylfuran was found in five pear cultivars and was perceived as having a fruity, green, earthy and vegetable-like smell [22]. Benzonitrile which was identified in *Pyrus ussuriensis* cultivars [4] was also indentified in longyuanyangli and D Jules. Other compounds such as phenol, (*Z*)-rose oxide and (*E*)-rose oxide were also detected in this study. Various aroma components and concentration difference determine the overall flavor properties of pears.

### 2.4. Cluster Analysis (CA)

The cluster analysis based on concentrations of identified volatile compounds was performed using the SPSS Statistical 19.0 software. The dendrogram (Figure 5) shows that two main groups are distinguished. Longyuanyangli, which has the maximum aroma numbers and the highest concentrations, is separated from other pear cultivars. Packham, D Jules, Clapp and Stark constitute the second group. They are four Occidental pears, which are introduced from abroad. Figure 5 shows that the D Jules and Stark have the slightest differences compared with other cultivars. Many factors affect the volatile compounds composition of the fruits. In this study, the volatile compositions of pears were found to be considerably different.

## 3. Materials and Methods

### 3.1. Materials

Five pear cultivars were prepared for analysis. The detailed information of the pears is shown in Table 1. The conventional indicators such as growing period, external morphology and skin color were used to judge the maturity of each cultivar. All fruits were stored at 1 °C before experiments. For each cultivar in this study, after-ripening process was necessary to enhance the flavor and taste. Samples were placed at room temperature before experiments (approximately five days). The 4 SPME fibres (100 μm PDMS, 85 μm PA, 65 μm PDMS/PVB and 50/30 μm DVB/CAR/PDMS) were supplied by Supelco (Bellefonte, PA, USA). The length of the fibre coating is 1 cm. The internal standard 2-nonanone (>99%) was obtained from Dr. Ehrensorfer (Germany).

### 3.2. Volatiles Extraction

HS-SPME was used for volatile compounds extraction. A 50/30 μm DVB/CAR/PDMS SPME fibre was used in this study. Fibres were activated according to the conditioning guidelines before the first use. The core of each pear was removed, while the peel was reserved. The skin and flesh of each pear was cut into cubes (0.5 cm × 0.5 cm × 0.5 cm). For each extraction, 6.0 g of sample were placed into a 15-mL screw-cap vial. Prior to sealing of the vials, 5 μL of 10 μg/mL 2-nonanone was added as internal standard, and was standing for 10 min. Then each vial was placed in a constant-temperature controller at 40 °C for 40 min. Finally, the SPME fibre was immediately inserted into the GC injector for desorption at 270 °C for 2 min in the split mode of 10:1.

### 3.3. GC×GC-TOFMS Conditions

The volatile compounds analysis was performed with an Agilent 7890B gas chromatography equipped with a Pegasus 4D-C time-of-flight mass spectrometric detector. A Rxi-5MS column (30 m × 250 μm × 0.25 μm) was used as the first-dimension (1D) column, and a Rxi-17Sil MS column (2 m × 250 μm × 0.25 μm) was used as the second-dimension (2D) column. Helium was used as the carrier gas at a constant flow of 1.4 mL/min. The front inlet and the transfer line temperature were 270 °C and 280 °C, respectively. The oven temperature programme conditions were as follows: initial temperature was 40 °C for 2 min, rose at 5 °C/min up to 200 °C, then ramped to 280 °C at 20 °C/min and hold for 2 min. The secondary oven temperature was kept at 5 °C above the GC oven temperature throughout the chromatographic run. The modulator temperature was offset by 15 °C in relative to the secondary oven temperature. The modulation period was set for 3 s with a 0.6 s hot pulse time.

The MS parameters were as follows: acquisition delay 60 s, acquisition rate 100 (spectra/s), the acquisition voltage 1450 V, electron energy −70 V, ion source 250 °C. Mass spectra were acquired in the *m*/*z* range 35–550 amu.

### 3.4. Data Processing and Statistical Analysis

LECO ChromaTOF Version 4.73.3.0 software (Leco Corporation, St. Joseph, MO, USA) was used for instrument control, data acquisition and processing. Each chromatograph peak was compared to National Institute of Standards and Technology (NIST2017) library and the minimum similarity is 800. The area of the base peak was used for quantification. The quantitative analysis for aroma components was carried out by internal standard method using 2-nonanone as an internal standard. Therefore, the concentration of each volatile compound was normalized to that of 2-nonanone. The formula for volatile compounds quantification is as follows:
$$C_{a}=\frac{\frac{PA_{a}}{PA_{is}}\times C_{is}\times 5\;\mu L}{m}$$
*C_a_*: the concentration of aroma components (ng⁄g); *PA_a_*: peak area of aroma components; *PA_is_*: peak area of internal standard; *C_is_*: the concentration of internal standard (g⁄mL); *m*: mass of sample (g). The concentration of the 2-nonanone was 10 μg/mL and the mass of sample was 6.0 g. Data are means ± SD of three replications. Cluster analysis (CA) was performed using the SPSS Statistical 19.0.

## 4. Conclusions

In conclusion, the combination of SPME and GC×GC-TOFMS has improved the analysis of pear volatile compounds. The 50/30 μm DVB/CAR/PDMS SPME fibre exhibited obvious advantages for volatile compounds extraction. A total of 241 compounds were identified in five pear cultivars, which are primarily esters, alcohols, aldehydes and alkenes. Volatile compounds such as sesquirosefuran and anethole are reported for the first time in pears. Evaluation of aromas at the germplasm level will facilitate breeding efforts and improve sensory quality of fruits. This research will contribute to further studies related to volatile compounds analysis.

## Figures and Tables

**Figure 1 molecules-24-01795-f001:**
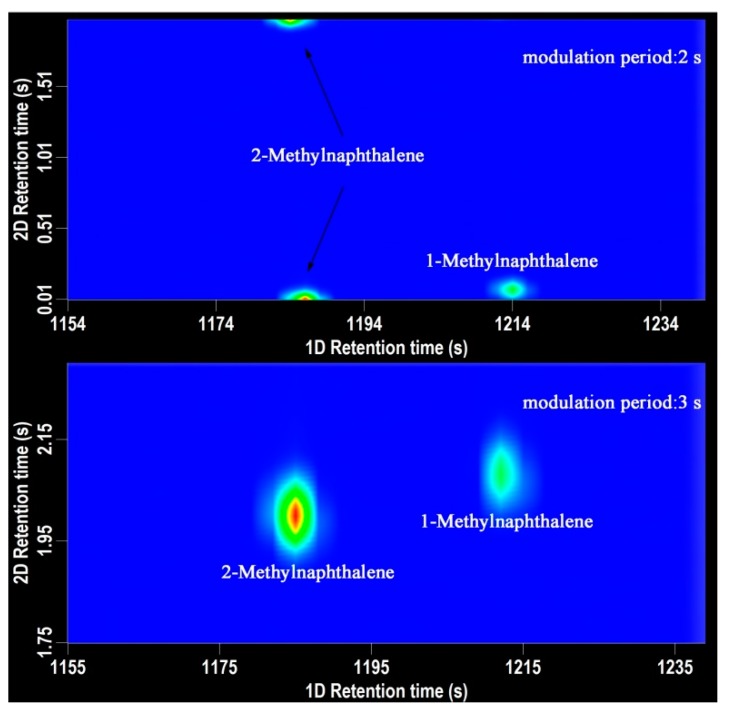
The 2D chromatography of 2-methylnaphthalene and 1-methylnaphthalene. (top-modulation period 2 s, bottom-modulation period 3 s).

**Figure 2 molecules-24-01795-f002:**
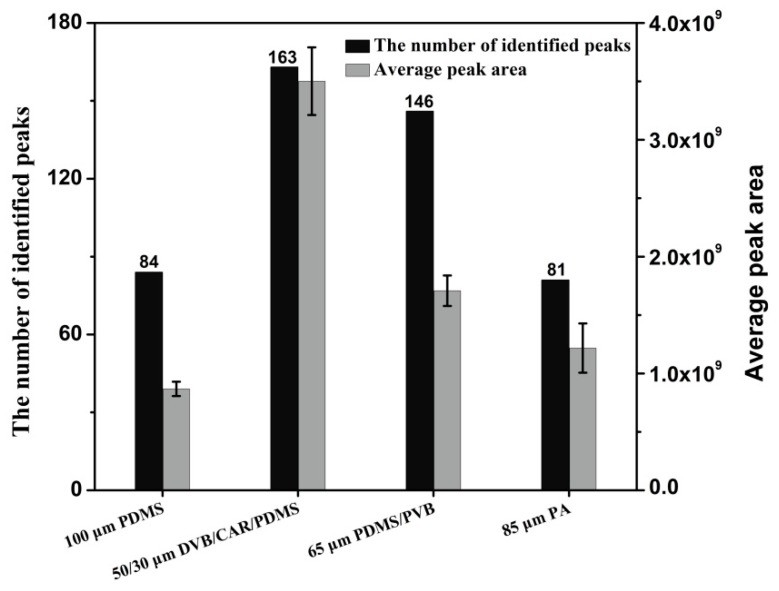
Comparison of aroma amounts and peak areas in pears using four different SPME fibres.

**Figure 3 molecules-24-01795-f003:**
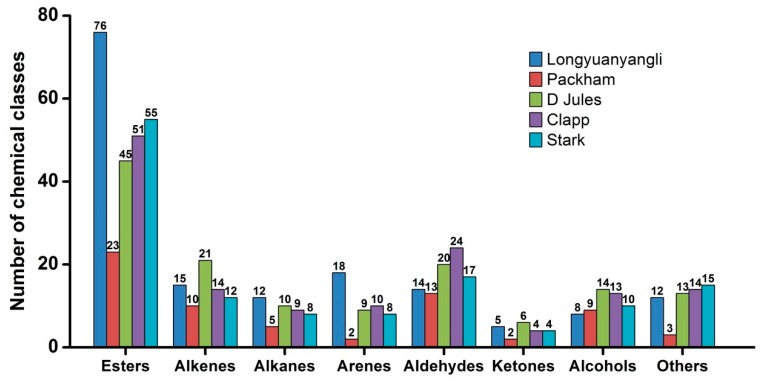
The number of chemical classes in pears.

**Figure 4 molecules-24-01795-f004:**
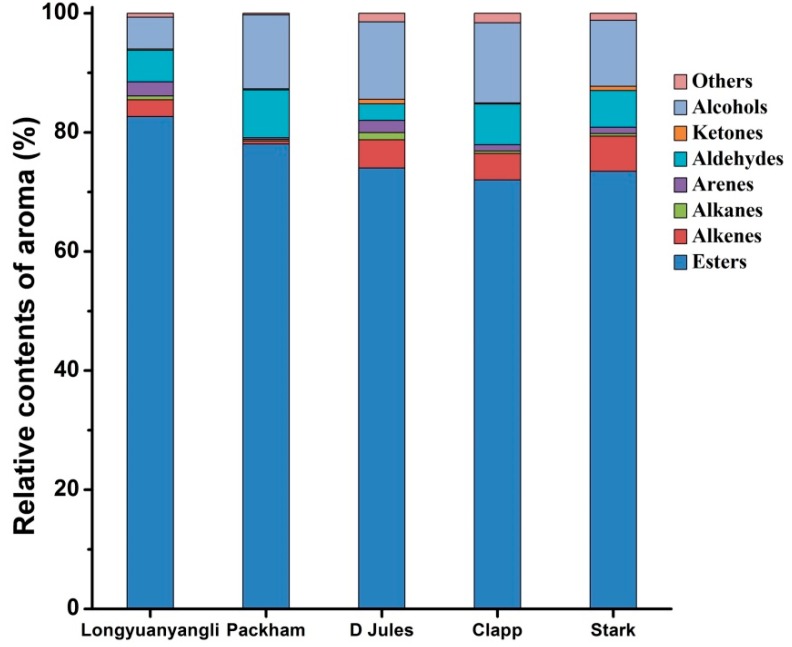
Relative contents of volatile compounds in five pears.

**Figure 5 molecules-24-01795-f005:**
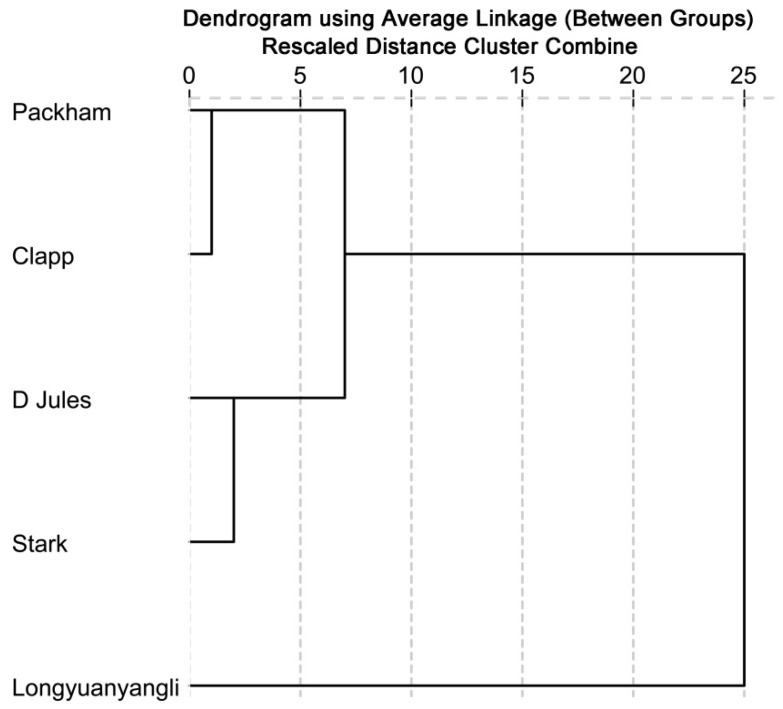
Dendrogram obtained from cluster analysis based on the identified volatile compounds.

**Table 1 molecules-24-01795-t001:** The cultivars, abbreviation, producing area and sampling time of 5 pears.

Pear Cultivars	Abbreviation	Producing Area	Sampling Time
Longyuanyangli	longyuanyangli	Qiqihaer city of Heilongjiang province	07 September 2018
Packham’s Triumph	Packham	Weihai city of Shandong Province	27 September 2018
Docteur Jules Guyot	D Jules	Yantai city of Shandong Province	01 August 2018
Clapp’s Favorite	Clapp	Yantai city of Shandong Province	07 August 2018
Starkrimson	Stark	Yantai city of Shandong Province	01 August 2018

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
