# Peer review of "Analysis of Volatile Compounds in Pears by HS-SPME-GC×GC-TOFMS"

_molecules, 2019, doi:10.3390/molecules24091795_

Round 1
Reviewer 1 Report
The manuscript wirtten by Wang et al. presents new results from a GCxGC-TOF experiment of 5 different pear cultivars. The results are well presented and well discussed with extensive references to previous litterature, shading light on the originality of the work proposed.
All the recommendations I made in a previous review have been answered to. I recommend then publication of this work.
I noticed a few sentences that could be clarified or typos :
l.46 suppress "always"
l.55 Are all the pears of the same type (P. communis ?) ?
l.82 Why yali pear to test SPME fibers ? How do it compare to the other cultivars ?
l.107-108 : you use Ia for the calculated retention index in the equation whereas it is noted RIc in table 1. Please be consistent.
l.233 Figure 5 shows that the D Jules and Stark have the slight differences compared with other cultivars" => "slightest" ?
Author Response
Journal: Molecules
Manuscript ID: molecules-495955
Type: Article
Title: "Analysis of volatile compounds in pears by HS-SPME-GC×GC-TOFMS"
Correspondence Author: Zi-lei Chen ([email protected]) and Guo-sheng Yang ([email protected])
#Reviewer 1
Thank you for your comments. The answers to the comments are marked in bold below. Revised portion are marked in red in the manuscript. The corrections in the paper and the responds to the reviewer’s comments are as following:
1. l.46 suppress "always"
Answer: The word "always" was deleted in l.46.
2. l.55 Are all the pears of the same type (P. communis ?) ?
Answer: Packham, D Jules, Clapp and Stark are 4 Occidental pears which are introduced from abroad. Longyuanyangli is a hybrid variety. Some research attributed longyuanyangli to the P. communis system according to its characteristics and intense aroma. But we cannot find strong evidence to confirm this problem. We have re-written this sentence to make the expression clear.
“Longyuanyangli is a hybrid variety that has intense aroma.”
3. l.82 Why yali pear to test SPME fibers ? How do it compare to the other cultivars ?
Answer: In order to find a optimum SPME fibre, we should use a pear that have strong aroma. Yali pear was selected because its strong aroma. Compare to other cultivars used in this study, the aroma of yali pear smells more intense without consideration of detection result.
4. l.107-108 : you use Ia for the calculated retention index in the equation whereas it is noted RIc in table 1. Please be consistent.
Answer: The “Ia” was replaced by “RIa” in the equation. RIa: the retention index of the compound of interest; a: the compound of interest. Moreover, “RIc” was replaced by “RI” in table 1.
5. l.233 Figure 5 shows that the D Jules and Stark have the slight differences compared with other cultivars" => "slightest" ?
Answer: The word “slight” was replaced by "slightest".

Reviewer 2 Report
The authors present a method to improve assessment of the volatile compounds from pears. While interesting, it suffers from some difficulties in writing, data presentation, and explanation outlined below.
Major points
I started reading the paper and was excited about the methodology. Unfortunately, the grammar and word choice made reading and assessing the work quite difficult because they were so distracting and occasionally obscured the meaning. Articles are misused or absent, comma use is all over the place, and words are written that do not mean what the authors are trying to communicate [I hope]. I started editing the writing in the intro, but stopped within the first two paragraphs because it was going to become an overly involved task. It is necessary to edit the work with a keen eye for English grammar/structure/syntax/word use.
Preparation of the figures is sloppy. Figure 1 doesn’t even have axis labels! The font sizes are different between figures (some of the fonts might also be different), some have axis labels with all nouns capitalized, the first noun capitalized, or none of the nouns capitalized. There is no consistent color scheme for the axis labels, and the error bars aren’t even mentioned (are they SD or SEM? How many replicates?). Taken a step further, if the data was gathered in triplicate, why do they not state that they are the mean +/- SD (as an aside, why did I have to search for this information buried in the M&M?)? Are you implying that your data gathering is so perfect that every single time you tested you saw every single compound even if it was a minor constituent? If so, congratulations and state it, but I find this result unlikely considering it is a biological system with inherent noise and a non-homogenous distribution of components within fruit flesh. Furthermore, Table 1 is a lovely and hulking collection of data that really belongs in the supplemental material with a summary table in the manuscript, or simply rely on the figures to convey the trends of the data. In general, the data presentation leaves much to be desired and must be improved before publication.
Retention index is used as part of the identification of the compounds, which is entirely appropriate in this paper. That said, how do you know that your system was working? You should be running a retention index standard set regularly, and there is no mention of this anywhere in the manuscript. This must be done to account for variations across systems and within a system over time.
In the intro, you state that SPME is the way that groups have assessed pear volatiles, and then present data on the concentration of compounds from your pears as compared to the 2-nonanone standard. Why was 2-nonanone used as the internal standard? No logic or reason was presented as to this choice of standard. What compound(s) do previous papers on pear volatiles use as their semi-quantitative compound standard? Has quantification been done on other pears and are your results in any way comparable to previous work, even within the same order of magnitude? The discussion largely ignores any comparison of the quantitative aspects of this manuscript with those that have preceded it. That should be remedied or a convincing reason should be presented to account for its absence.
You do cluster analysis and then do a rather cursory job of describing the result. What groups of compounds account for the clustering? What compound classes are the major drivers of similarity and divergence? Is this more based on number of compounds, ratios of compounds to amounts, or does the amount of compound hold more weight in this analysis? While some sort of analysis like this or PCA is usually included, the results need to be explained in sufficient detail to justify the presence of the graph. This could be an interesting result, but the reasons underlying/explaining the clustering behavior are absent, which means that this result cannot be compared to other systems.
Minor points
Line 27: “one of the most popular fruit crops” – where? In the world? Says who? This is entirely subjective and not substantiated without citations. Either soften or cite.
Line 32: This suffers from the same problem as line 27 – a sweeping general claim without evidence, data, or anything other form of support. Either soften or cite.
Line 33: Researchers is misspelled. Also, if you state that work has been done, why is not cited there? You can’t claim work has been done and not cite it.
Line 37: Conditions shouldn’t be plural as written.
Line 40: “the external factors” – use of “the” indicates that this list is exhaustive and is likely incorrect with “often” as written. Remove “the” here and assess article usage throughout because there many more instances of incorrect articles in the manuscript.
Line 41: Comprise is probably more accurately include as written. Also on this line, no, water is the main component of pears, not esters. Do you mean volatile/aroma compounds? This sentence includes the words “occupy main aspects,” which do not have a meaning as written.
Line 45: Always – don’t use exclusive language unless it is correct – do you mean to write that another method has never been used? How was work done before SPME?
Line 175: Why are words red? Are they special, or did you mean to change them before submission and then forgot?
Line 256: Do you mean a NIST similarity of 80%?
Line 267: What was vast about the improvement in this paper? There is no comparison between these results and the quantitative results from other studies. This makes the use of “vast” fairly subjective.
Supplemental: Add axis titles to the graphs.
Author Response
Journal: Molecules
Manuscript ID: molecules-495955
Type: Article
Title: "Analysis of volatile compounds in pears by HS-SPME-GC×GC-TOFMS"
Correspondence Author: Zi-lei Chen ([email protected]) and Guo-sheng Yang ([email protected])
#Reviewer 2
Thanks for your comments concerning our manuscript. Those comments are all valuable and very helpful for revising and improving our paper, as well as the important guiding significance to our researches. We have studied comments carefully and have made correction which we hope meet with approval. The comments are marked in bold below and the answers are near to the comment.
Revised portion are marked in red in the manuscript. The main corrections in the paper and the responds to the reviewer’s comments are as following:
# Major points
1. I started reading the paper and was excited about the methodology. Unfortunately, the grammar and word choice made reading and assessing the work quite difficult because they were so distracting and occasionally obscured the meaning. Articles are misused or absent, comma use is all over the place, and words are written that do not mean what the authors are trying to communicate [I hope]. I started editing the writing in the intro, but stopped within the first two paragraphs because it was going to become an overly involved task. It is necessary to edit the work with a keen eye for English grammar/structure/syntax/word use.
Answer: we have modified the English language in our manuscript according to your comments. The detail was shown in manuscript in red font.
2. Preparation of the figures is sloppy. Figure 1 doesn’t even have axis labels! The font sizes are different between figures (some of the fonts might also be different), some have axis labels with all nouns capitalized, the first noun capitalized, or none of the nouns capitalized. There is no consistent color scheme for the axis labels, and the error bars aren’t even mentioned (are they SD or SEM? How many replicates?). Taken a step further, if the data was gathered in triplicate, why do they not state that they are the mean +/- SD (as an aside, why did I have to search for this information buried in the M&M?)? Are you implying that your data gathering is so perfect that every single time you tested you saw every single compound even if it was a minor constituent? If so, congratulations and state it, but I find this result unlikely considering it is a biological system with inherent noise and a non-homogenous distribution of components within fruit flesh. Furthermore, Table 1 is a lovely and hulking collection of data that really belongs in the supplemental material with a summary table in the manuscript, or simply rely on the figures to convey the trends of the data. In general, the data presentation leaves much to be desired and must be improved before publication.
Answer: ①The axis labels of figure 1 were added.
②The axis labels of figures were all capitalized by the first noun.
③The fonts were same in the figures. If font size such as numerical labels in figure 3 becomes bigger, the font will overlap with the image.
④The color of the axis labels of figure 2 was consistent.
⑤According to your comment, table 1 was put in the supplemental material (table S2).
⑥we were not to that the data gathering is so perfect that every single time we can saw every single compound even if it was a minor constituent. The sentence “The aroma extraction process was repeated for three times to guarantee the accuracy of the results” was put in section 2.2. Many volatile compounds were obtained at each repetition. For each aroma, each repetition produced a matching factor with the NIST. When the similarity of each volatile compound higher than 800, the volatile compound was confirmed. For a aroma component, when this aroma component was detected in all three replications and when the similarities of three replications are all above 800, this component is confirmed. Therefore, for a minor constituent, if it only appears within two repetitions, it will be deleted.
3. Retention index is used as part of the identification of the compounds, which is entirely appropriate in this paper. That said, how do you know that your system was working? You should be running a retention index standard set regularly, and there is no mention of this anywhere in the manuscript. This must be done to account for variations across systems and within a system over time.
Answer: Under GC conditions described in section 3.3, the retention time of C5-C20 were obtained and the retention index of volatile compounds was calculated. The calculated retention index of each volatile compound was contrasted with the papers. Therefore, each volatile compound in this study was verified according to papers. The retention time of C5-C20 was as follows.
Table 1 The retention time of C5-C20 and the given retention index
Alkanes | Formula | R.T. (s) | CAS | Retention Index |
Pentane | C5H12 | 102, 0.520 | 109-66-0 | 500 |
n-Hexane | C6H14 | 129, 0.580 | 110-54-3 | 600 |
Heptane | C7H16 | 192, 0.650 | 142-82-5 | 700 |
Octane | C8H18 | 309,0.750 | 111-65-9 | 800 |
Nonane | C9H20 | 477,0.820 | 111-84-2 | 900 |
Decane | C10H22 | 660,0.870 | 124-18-5 | 1000 |
Undecane | C11H24 | 846,0.920 | 1120-21-4 | 1100 |
Dodecane | C12H26 | 1023,0.960 | 112-40-3 | 1200 |
Tridecane | C13H28 | 1191,0.980 | 629-50-5 | 1300 |
Tetradecane | C14H30 | 1347,1.000 | 629-59-4 | 1400 |
Pentadecane | C15H32 | 1497.1010 | 629-62-9 | 1500 |
Hexadecane | C16H34 | 1638,1.030 | 544-76-3 | 1600 |
Heptadecane | C17H36 | 1773,1.040 | 629-78-7 | 1700 |
Octadecane | C18H38 | 1899,1.050 | 593-45-3 | 1800 |
Nonadecane | C19H40 | 2019,1.060 | 629-92-5 | 1900 |
Eicosane | C20H42 | 2118,0.790 | 112-95-8 | 2000 |
The ChromaTOF-GC uses the Van den Dool and Kratz equation for the calculation of the retention index. The equation is:
Where:
Ia = the retention index of the compound of interest.
a = the compound of interest.
n = the carbon number of the lower normal alkane.
N = the carbon number of the higher normal alkane.
RT = the retention time.
The retention index do not changed except for the retention time of alkanes are changed. Therefore, under the same experimental conditions, the retention index is consistent. Moreover, same alkanes also existed in pear aroma, such as undecane, dodecane, tridecane and tetradecane. We can compare the calculated alkane retention index with the retention index of the alkane setted in Table 1.
4. In the intro, you state that SPME is the way that groups have assessed pear volatiles, and then present data on the concentration of compounds from your pears as compared to the 2-nonanone standard. Why was 2-nonanone used as the internal standard? No logic or reason was presented as to this choice of standard. What compound(s) do previous papers on pear volatiles use as their semi-quantitative compound standard? Has quantification been done on other pears and are your results in any way comparable to previous work, even within the same order of magnitude? The discussion largely ignores any comparison of the quantitative aspects of this manuscript with those that have preceded it. That should be remedied or a convincing reason should be presented to account for its absence.
Answer: According to references 3 and 4, the internal standard 3-nonanone was used. Before quantitative calculation, the aroma of occidental pear (pyrus communis L.) cultivars was assessed without the internal standard. Volatile compounds of pears comprise esters, aldehydes, alcohols, ketones and hydrocarbons. Therefore, we should find a substance that does not belong to the aroma of pears as an internal standard. 2-nonanone was selected in this study. Relative concentrations have been calculated referring to an internal standard of 2-nonanone. Therefore, in this manuscript, the concentrations all refer to relative concentration. The results were obtained under different experiment conditions and various concentrations of internal standard. For pear aroma analysis, the comparison of the quantitative aspects of this manuscript with other papers makes no sense.
[3] Chen, Y.Y.; Yin, H.; Wu, X.; Shi, X.J.; Qi, K.J.; Zhang, S.L. Comparative analysis of the volatile organic compounds in mature fruits of 12 Occidental pear (Pyrus communis L.) cultivars. Sci. Hortic. 2018, 240, 239-248.
[4] Qin, G.H.; Tao, S.T.; Cao, Y. F.; Wu, J.Y.; Zhang, H.P.; Huang, W.J.; Zhang, S.L. Evaluation of the volatile profile of 33 Pyrus ussuriensis cultivars by HS-SPME with GC-MS. Food Chem. 2012, 134, 2367-2382.
5. You do cluster analysis and then do a rather cursory job of describing the result. What groups of compounds account for the clustering? What compound classes are the major drivers of similarity and divergence? Is this more based on number of compounds, ratios of compounds to amounts, or does the amount of compound hold more weight in this analysis? While some sort of analysis like this or PCA is usually included, the results need to be explained in sufficient detail to justify the presence of the graph. This could be an interesting result, but the reasons underlying/explaining the clustering behavior are absent, which means that this result cannot be compared to other systems.
Answer: In our previous work, the analysis about PCA was carried out, but there were no real conclusions on the results obtained except for a brief description of the results. We tried to draw an explicit conclusion like reference 3 and 4, but the numbers of the volatile compounds are so many that we cannot distinguish correctly. Therefore, according to reference 16 and 31, the cluster analysis was performed. Cluster analysis is a complex and diversified statistical method that has a complicated calculation process. In this study, the cluster analysis based on concentrations of identified volatile compounds was performed. According to reference 16 and 31, the same analysis was obtained.
[3] Chen, Y.Y.; Yin, H.; Wu, X.; Shi, X.J.; Qi, K.J.; Zhang, S.L. Comparative analysis of the volatile organic compounds in mature fruits of 12 Occidental pear (Pyrus communis L.) cultivars. Sci. Hortic. 2018, 240, 239-248.
[4] Qin, G.H.; Tao, S.T.; Cao, Y.F.; Wu, J.Y.; Zhang, H.P.; Huang, W.J.; Zhang, S.L. Evaluation of the volatile profile of 33 Pyrus ussuriensis cultivars by HS-SPME with GC-MS. Food Chem. 2012, 134, (4), 2367-2382.
[16] Gokbulut, I.; Karabulut, I. SPME-GC-MS detection of volatile compounds in apricot varieties. Food Chem. 2012, 132, (2), 1098-1102.
[31] Zakaria, S.R.; Saim, N.; Osman, R.; Abdul Haiyee, Z.; Juahir, H. Combination of sensory, chromatographic, and chemometrics analysis of volatile organic compounds for the discrimination of authentic and unauthentic harumanis mangoes. Molecules, 2018, 23, 2365.
# Minor points
1. Line 27: “one of the most popular fruit crops” – where? In the world? Says who? This is entirely subjective and not substantiated without citations. Either soften or cite.
Answer: We have re-written this sentence “Pear (Pyrus spp., Rosaceae) is a popular fruit and is extensively grown in China.”
2. Line 32: This suffers from the same problem as line 27 – a sweeping general claim without evidence, data, or anything other form of support. Either soften or cite.
Answer: Thanks for your comment. The reference 2 was added in line 32.
[2] Rapparini, F; Predieris, S. Pear fruit volatiles. Hort. Rev. 2003, 28, 237-324.
3. Line 33: Researchers is misspelled. Also, if you state that work has been done, why is not cited there? You can’t claim work has been done and not cite it.
Answer: in this sentence, the word “researches” is the plural form of “research”. The word “researches” was replaced by “studies” to make the expression clear. The reference 2-7 was added.
4. Line 37: Conditions shouldn’t be plural as written.
Answer: L 37: “Investigations” was replaced by “Investigation” and “have” was replaced by “has”
L 39: “calcium treatments” was replaced by “calcium treatment”
5. Line 40: “the external factors” – use of “the” indicates that this list is exhaustive and is likely incorrect with “often” as written. Remove “the” here and assess article usage throughout because there many more instances of incorrect articles in the manuscript.
Answer: “the” was deleted in L40. “often” was deleted in L40.
6. Line 41: Comprise is probably more accurately include as written. Also on this line, no, water is the main component of pears, not esters. Do you mean volatile/aroma compounds? This sentence includes the words “occupy main aspects,” which do not have a meaning as written.
Answer: We are very sorry for our unclear expression. We mean that esters are the major components of volatile compounds. We have re-written this sentence to make the expression clear. “Comprise” was replaced by “include”. The sentence is “Esters are the major volatile components in P. ussuriensis and P. communis, while in P. pyrifolia aldehydes are the dominant volatile compounds, followed by alcohols and esters.”
7. Line 45: Always – don’t use exclusive language unless it is correct – do you mean to write that another method has never been used? How was work done before SPME?
Answer: Thanks for your comments. We learned that methods such as direct solvent extraction, distillation, simultaneous distillation-extraction (SDE) and supercritical fluid extraction (SFE) had been applied in isolation of volatiles from pears. Since 1994, SPME has been applied in fruit flavor studies. The word "always" was deleted in l.45.
[2]. Rapparini, F; Predieris, S. Pear fruit volatiles. Hort. Rev. 2003, 28, 237-324.
8. Line 175: Why are words red? Are they special, or did you mean to change them before submission and then forgot?
Answer: This is a manuscript after revision and resubmission. According to the requirements of editors, all changes should be highlighted in manuscript. Therefore, the font of the changed content is displayed in red.
9. Line 256: Do you mean a NIST similarity of 80%?
Answer: The value 800 is a way of expression in similarity. It can be interpreted as 80%. According to reference 1 and 2, the similarity was expressed as a percentage figure. Nevertheless, in the reference 3, the similarity was expressed as three-digit numbers. In our instrument parameter, the similarity was expressed as three-digit numbers (the highest number is 999).
[1] Welke, J.E.; Zanus, M.; Lazzarotto, M.; Pulgati, F.H.; Zini, C.A. Main differences between volatiles of sparkling and base wines accessed through comprehensive two dimensional gas chromatography with time-of-flight mass spectrometric detection and chemometric tools. Food Chem. 2014, 164, 427-437.
[2] Vestner, J.; Malherbe, S.; Toit, M.D.; Nieuwoudt, H.H.; Mostafa, A.; Gòrecki, T.; Tredoux, A.G.J.; Villiers, A. Investigation of the Volatile Composition of Pinotage Wines Fermented with Different Malolactic Starter Cultures Using Comprehensive Two-Dimensional Gas Chromatography Coupled to Time-of-Flight Mass Spectrometry (GC×GC-TOFMS). J. Agric. Food Chem. 2011, 59, 12732–12744.
[3] Magagna, F.; Liberto, E.; Reichenbach, S.E.; Tao, Q.P.; Carretta, A.; Cobelli, L.; Giardina, M.; Bicchi, C.; Cordero, C. Advanced fingerprinting of high-quality cocoa: Challenges in transferring methods from thermal to differential-flow modulated comprehensive two dimensional gas chromatography. J. Chromatogr. A, 2018, 1536, 122-136.
10. Line 267: What was vast about the improvement in this paper? There is no comparison between these results and the quantitative results from other studies. This makes the use of “vast” fairly subjective.
Answer: “vastly” was deleted. For pear aroma analysis, the comparison of the quantitative aspects of this manuscript with other papers makes no sense. The main improvement of this study is to obtain a variety of aroma components.
11. Supplemental: Add axis titles to the graphs.
Answer: The axis titles were added in the graphs.

Round 2
Reviewer 2 Report
The authors did a thorough job answering the comments and I feel that the presentation of the study has improved. I would make two final suggestions for completeness:
The authors provide a detailed response to the critique, but then do not alter the M&M section of the manuscript accordingly. I would modify the sentence on line 233 as follows:
"Prior to sealing of the vials, 5 μL of 10 μg/mL 2-nonanone, chosen for use as internal standard because of its inclusion in previous experiments [two citations], was added and the vials were left standing for 10 min."
Add a sentence to section 3.4 regarding your use of standards to determine the RI as this is fair pertinent to your methods.
This manuscript is a resubmission of an earlier submission. The following is a list of the peer review reports and author responses from that submission.
Round 1
Reviewer 1 Report
The manuscript called "Analysis of volatile compounds in pears by HS-SPME-GCxGC-TOFMS" by Wang et al. describe aroma molecular content analyzed by a SPME technique on pear. 5 pear cultivars are studied. A long and well described list of chemical compounds is established by the authors, among them compounds that were not previsouly observed by other studies. The work is of interest and well conducted, and written. The methodology is well described and the results are correctly exposed and discussed.However, I have one main concern concerning the principal component analysis. The discussion of results is generally well conducted except for this last part. It is not clear what kind of variability is accounted in the PCA : diversity? concentration? all pears are used ? It is then difficult to interprete the results and to evaluate the goal of such complex analysis.
PCA is an interesting tool and the effort to include such analysis is appreciated, but the description is difficult to understand for the reader, as apart from a brief description of the results there is no real conclusions on the results obtained. For instance, I would have appreciate to read a description of the goal of the analysis, and the type of data used.
Moreover, have you tried to compare results from the 5 pear cultivars individually in this analysis so as to identify the specificity of each family ?
The PCA adds a real value to the work, but it should be thorougly rewritten to be more explicit about the goal, the method and the conclusions.
I also have some minor remarks:
1) Concerning SPME fiber selection: The fibers used are different materials. They should promote different kind of family of compounds. Have you checked that possibility ? What type of compounds you would expect from each one and is one of the fiber more sensitive to a particular compound than the 50/30 µm DVB/CAR/PDMS one ? To answer that question, you could add an argumentation on the chemical diversity (if there is) observed between those fibers.
2) There is a great variability of ethanol content between samples (SD is quite elevated) (l.121-122). Are the aging and treatment of pear different ? If known could you add some information on that matter (in the materials section for instance) ?
3 ) Typos and remarks:
l.31 "five" > "fifth"
l.70-71 "for 3s with a 0.6s hot pulse time" Is it 3+0.6 or 3 and 0.6 included ?
l.146 "were existed" > "existed"
l.151 "even their contents" > "even if their contents". "than others" > "than other compounds"
l.161 "contributing woody smell"> "contributing to the woody smell"
l.165 "In total" > "A total"
l.215-229 : Are those conditions typical for that fiber and GCxGC. Could you add a reference if necessary.
l.233 What does the value "800" accounts for ? The unity/value used is not standard. Is it 80% ?
l.237-238 Equation would be clearer if you used litterals value such as Ci or P_IS, and describe them in the following lines.
Author Response
Journal: Molecules
Manuscript ID: molecules-435985
Type: Article
Title: "Analysis of volatile compounds in pears by HS-SPME-GC×GC-TOFMS"
Correspondence Author: Zi-lei Chen ([email protected]) and Guo-sheng Yang ([email protected])
Thank you for your comments concerning our manuscript. Those comments are all valuable and very helpful for revising and improving our paper, as well as the important guiding significance to our researches. We have studied comments carefully and have made correction which we hope meet with approval. The answers to the comments are marked in bold below. Revised portion are marked in red in the manuscript. Minor corrections were marked in red in manuscript. The main corrections in the paper and the responds to the reviewer’s comments are as following:
#Reviewer 1
1. It is not clear what kind of variability is accounted in the PCA : diversity? concentration? all pears are used ? It is then difficult to interprete the results and to evaluate the goal of such complex analysis. Moreover, have you tried to compare results from the 5 pear cultivars individually in this analysis so as to identify the specificity of each family?
Answer: We have re-written this part according to your suggestions. The PCA adds a real value to the work. In our previous work, there were no real conclusions on the results obtained except for a brief description of the results. We tried to draw an explicit conclusion like reference 3 and 4, but the numbers of the volatile compounds are so many that we can not distinguish correctly. Therefore, according to your comment and reference 16, the cluster analysis was performed.
[3] Chen, Y.Y.; Yin, H.; Wu, X.; Shi, X.J.; Qi, K.J.; Zhang, S.L. Comparative analysis of the volatile organic compounds in mature fruits of 12 Occidental pear (Pyrus communis L.) cultivars. Sci. Hortic. 2018, 240, 239-248.
[4] Qin, G.H.; Tao, S.T.; Cao, Y.F.; Wu, J.Y.; Zhang, H.P.; Huang, W.J.; Zhang, S.L. Evaluation of the volatile profile of 33 Pyrus ussuriensis cultivars by HS-SPME with GC-MS. Food Chem. 2012, 134, (4), 2367-2382.
[16] Gokbulut, I.; Karabulut, I. SPME-GC-MS detection of volatile compounds in apricot varieties. Food Chem. 2012, 132, (2), 1098-1102.
The cluster analysis and figure 5 were added in section 2.4 to clear expression the goal and the conclusion.
“The cluster analysis based on concentrations of identified volatile compounds was performed using the SPSS Statistical 19.0 software. The dendrogram (Figure 5) shows that two main groups are distinguished. Longyuanyangli, which has the maximum aroma numbers and the highest concentrations, is separated from other pear cultivars. Packham, D Jules, Clapp and Stark constitute the second group. They are 4 Occidental pears which are introduced from abroad. Figure 5 shows that the D Jules and Stark have the slight differences compared with other cultivars.”
Figure 5 Dendrogram obtained from cluster analysis based on the identified volatile compounds.
2. Concerning SPME fiber selection.
Answer: Thanks your suggestions. We think that your comments are very useful to our work. The number of chemical classes of each pear was identified.
50/30 μm DVB/CAR/PDMS | 85 μm PA | 65 μm PDMS/PVB | 100 μm PDMS | |
Esters | 67 | 49 | 62 | 46 |
Alkenes | 19 | 3 | 27 | 12 |
Alkanes | 19 | 4 | 17 | 14 |
Arenes | 20 | 2 | 15 | 0 |
Aldehydes | 9 | 6 | 10 | 2 |
Ketones | 5 | 2 | 4 | 4 |
Alcohols | 9 | 7 | 7 | 4 |
Others | 15 | 8 | 4 | 2 |
Total | 163 | 81 | 146 | 84 |
Phenol and isopropenylphenol were identified using the 85 μm PA fibre. Considering the peak numbers and the average peak areas for volatile compounds, the 50/30 μm DVB/CAR/PDMS fibre was selected for aroma extraction in this study. When special type of compounds was analyzed, we think that we can compare the differences between fibres.
3. There is a great variability of ethanol content between samples (SD is quite elevated) (l.121-122). Are the aging and treatment of pear different? If known could you add some information on that matter (in the materials section for instance) ?
Answer: Thanks for your suggestions. The aging and the treatment of pear were same. In GC×GC-TOFMS, when the number of carbon is less than 4, such as ethanol and acetaldehyde, the substances cannot focus in modulator. Therefore, sometimes the SD is quite elevated.
4. l.70-71 "for 3s with a 0.6s hot pulse time" Is it 3+0.6 or 3 and 0.6 included ?
Answer: Modulation period is a significant parameter in GC×GC analysis. The hot pulse time and cold pulse time make up the modulation period together. So in this study, the modulation period was 3 s, and the hot pulse time and cold pulse time were 0.6 s and 0.9 s, respectively. Similar expression ‘The modulation period was set for 7 s with a 1.4 s hot pulse time’ is also existed in reference [20].
[20] Welke, J.E.; Zanus, M.; Lazzarotto, M.; Pulgati, F.H.; Zini, C.A. Main differences between volatiles of sparkling and base wines accessed through comprehensive two dimensional gas chromatography with time-of-flight mass spectrometric detection and chemometric tools. Food Chem. 2014, 164, 427-37.
5. l.215-229: Are those conditions typical for that fiber and GCxGC. Could you add a reference if necessary.
Answer: Thanks to your suggestion. The instrumental parameters including the modulation period, the oven temperature programme conditions and the acquisition voltage of the TOFMS were optimized in our study. Modulation period is a parameter of crucial importance in GC×GC-TOFMS analysis. So we emphatically discussed this parameter in our manuscript. Secondary oven temperature and modulator temperature are in accordance with reference [21]. The programmed heating rate of 5 °C/min and 8 °C/min was compared. The results show that the heating rate of 5 °C/min is suitable. The MS parameters included electron ionisation at 70 eV and ion source temperature at 250 °C, which was accordance with reference [21].
[21] Lebedev, A.T.; Polyakova, O.V.; Mazur, D.M.; Artaev, V.B.; Canet, I.; Lallement, A.; Vaitilingom, M.; Deguillaume, L.; Delort, A.M. Detection of semi-volatile compounds in cloud waters by GC×GC-TOF-MS. Evidence of phenols and phthalates as priority pollutants. Environ. Pollut. 2018, 241, 616-625.
6. l.233 What does the value "800" accounts for ? The unity/value used is not standard. Is it 80%?
Answer: The value 800 is a way of expression in similarity. It can be interpreted as 80%. According to reference 1 and 2, the similarity was expressed as a percentage figure. Nevertheless, in the reference 3, the similarity was expressed as three-digit numbers. In our instrument parameter, the similarity was expressed as three-digit numbers (the highest number is 999).
[1] Welke, J.E.; Zanus, M.; Lazzarotto, M.; Pulgati, F.H.; Zini, C.A. Main differences between volatiles of sparkling and base wines accessed through comprehensive two dimensional gas chromatography with time-of-flight mass spectrometric detection and chemometric tools. Food Chem. 2014, 164, 427-437.
[2] Vestner, J.; Malherbe, S.; Toit, M.D.; Nieuwoudt, H.H.; Mostafa, A.; Gòrecki, T.; Tredoux, A.G.J.; Villiers, A. Investigation of the Volatile Composition of Pinotage Wines Fermented with Different Malolactic Starter Cultures Using Comprehensive Two-Dimensional Gas Chromatography Coupled to Time-of-Flight Mass Spectrometry (GC×GC-TOFMS). J. Agric. Food Chem. 2011, 59, 12732–12744.
[3] Magagna, F.; Liberto, E.; Reichenbach, S.E.; Tao, Q.P.; Carretta, A.; Cobelli, L.; Giardina, M.; Bicchi, C.; Cordero, C. Advanced fingerprinting of high-quality cocoa: Challenges in transferring methods from thermal to differential-flow modulated comprehensive two dimensional gas chromatography. J. Chromatogr. A, 2018, 1536, 122-136.
7. l.237-238 Equation would be clearer if you used litterals value such as Ci or P_IS, and describe them in the following lines.
Answer: the equation was simplified as follows:
Ca: the concentration of aroma components (ng⁄g); PAa: peak area of aroma components; PAis: peak area of internal standard; Cis: the concentration of internal standard (g⁄mL); m: mass of sample (g).
Special thanks to you for your good comments.

Reviewer 2 Report
The manuscript entitled "Analysis of Volatile Compounds in Pears by HS-SPME-GC×GC-TOFMS" reports a screening of volatile compounds in 5 pears cultivars using SPME and GCxGC-ToF-MS. Different fibre coatings were compared for the extraction of volatile compounds.
The comparison between the 5 pears cultivar is biased by the sampling. Many factors affect the volatile compounds composition of the fruit: harvesting conditions, fruit ripening, storage only to cite few. Thus it is not possible to attribute the observed differences exclusively to the cultivar.
Extensive sampling is required. More fruits per varieties from different locations would reinforce the observed differences.
The data reported are not quantitative. Please refers to specific comment for details.
The identification of the compounds is not reliable. It is actually based exclusively on the MS spectra matching (with a score limit of 800). For further details refers to specific comments.
In the discussion of the results, the chemical class referring to terpenes (terpenes, terpenoids and sesquiterpenes) was split among the different chemical classes reported (L125-127; L152-163; L171-173) making the discussion of the results less coherent.
Specific comments:
L 21: change "was" in "were".
L 22-23: Compounds reported in the manuscript were not quantified by a calibration method. Relative concentrations have been calculated referring to an internal standard thus is not appropriate to refer to "aroma concentration" in ng/g. Referring to the "chromatographic area" or to the "relative concentration" sounds more appropriate.
L 31: the sentence is not clear. Please reformulate.
L 45: This is not true: volatile compounds in pear have been studied even before the introduction of the spme.
L 78-79 & L 202-203: please report the length of the coating.
L 80-84: It is not clear which samples were analysed for comparison between fibres: the same sample with all the fibres or the number of volatile compounds (and the areas) are from the combination of fibres with all pear cultivars?
L 84-86: was the DVB/CAR/PDMS fibre used to generate data in table 2? Please clarify.
L 90: the statement should be smoothed out: the "volatile compounds were tentatively identified". The only criterium used for identification was the ms spectra matching with NIST database (L 232-233). The use of retention index is necessary for the identification of the compounds. This is still more critical for the identification of terpenes and terpenoids that present very similar (if not identical) fragmentation pattern.
L 93: "classification" stand for "comparison"?
L 94 & fig. 4: it is not clear if data refers to the number of compounds identified or the concentration.
Table 2 should report the matching factor with the NIST and the retention times recorded for each compound as well as retention index at least for the 1st dimension.
L 141: 6-methyl-5-heptene-2-one can also originate from farnesene that is present in higher amount in the peel compared to the flesh.
L 184-198: It is not clear why PCA was performed and which is the usefulness for the discussion of the results. Which information provides?
L 201: description of sample preparation is limited. Were the fruits pealed? Which section (L 209) of the fruit was diced? How and how long were the fruits stored before the measurements?
L 201-202: Table 1 is reported after table 2. Information reported is not complete. Were the fruits comparable in terms of ripening? Were the fruits harvested and stored in comparable conditions?
L 210-211: Was the reproducibility of the method tested? if so, please reports the results for the 2-nonanone.
L 213-214: The split mode is un-usual for SPME, there is a reason for this choice?
L 228-229: "acquired" is more appropriate than "scanned" when referring to a ToF MS. Please correct it.
L 233: please provide NIST version.
Author Response
Journal: Molecules
Manuscript ID: molecules-435985
Type: Article
Title: "Analysis of volatile compounds in pears by HS-SPME-GC×GC-TOFMS"
Correspondence Author: Zi-lei Chen ([email protected]) and Guo-sheng Yang ([email protected])
#Reviewer 2
1. The comparison between the 5 pears cultivar is biased by the sampling. Many factors affect the volatile compounds composition of the fruit: harvesting conditions, fruit ripening, storage only to cite few. Thus it is not possible to attribute the observed differences exclusively to the cultivar.
Answer:Thanks for your suggestions. Many factors affect the volatile compounds composition of the fruit, such as, harvesting conditions, fruit ripening and storage conditions. The investigations with regard to ripening, harvest and storage conditions were carried out in other experiments. Although many factors affect volatile compounds composition, the cultivar plays an important role. Your suggestions will give guidance to our following work.
2. Extensive sampling is required. More fruits per varieties from different locations would reinforce the observed differences.
Answer:Thanks for your suggestions. We know your meaning that each variety can be obtained from different locations. And differences existed among different locations. Your suggestions will give guidance to our following work.
3. The data reported are not quantitative. Please refers to specific comment for details.
Answer:“Therefore, the concentration of each volatile compound was normalized to that of 2-nonanone.” was added in L 258 to explain the relative quantification. Therefore, in this manuscript, the concentrations all refer to relative concentration. Your suggestions will give guidance to our following work.
4. The identification of the compounds is not reliable. It is actually based exclusively on the MS spectra matching (with a score limit of 800). For further details refers to specific comments
Answer:In our study, to distinguish some similar aromas, the standards were used. For n-alkanes, which have the same fragment ions (i.e. m/z 43, 57, 71, 85, 99) at 70 eV, the extraction process of the standard for each alkane was performed to obtain their retention time. Therefore, the retention times of alkanes are identified. For terpenoids, we compare the retention times of many repetitions, they are consistent. Your comments can give guidance to our following work.
5. In the discussion of the results, the chemical class referring to terpenes (terpenes, terpenoids and sesquiterpenes) was split among the different chemical classes reported (L125-127; L152-163; L171-173) making the discussion of the results less coherent.
Answer: Thanks for your suggestions. We think that the terpene was a type of alkenes. Therefore, the terpenes were discussed in section 2.3.4. The terpene alcohols were discussed in section 2.3.2. If we changed the classifications in this study, the figures, tables and the structure of the manuscript will be changed significantly. We hope your understand.
Specific comments:
1. L 21: change "was" in "were".
Answer: We changed "was" to "were" in L 21.
2. L 22-23: Compounds reported in the manuscript were not quantified by a calibration method. Relative concentrations have been calculated referring to an internal standard thus is not appropriate to refer to "aroma concentration" in ng/g. Referring to the "chromatographic area" or to the "relative concentration" sounds more appropriate.
Answer: “Therefore, the concentration of each volatile compound was normalized to that of 2-nonanone.” was added in L258 to explain the relative quantification. Therefore, in this manuscript, the concentrations all refer to relative concentration.
L 22: the “relative concentration” was added to replace "aroma concentration"
L 113: the "aroma concentration" was replaced by “concentration’’
3. L 31: the sentence is not clear. Please reformulate.
Answer: We have re-written this sentence.
“The P. sinkiangensis Yu. was reported as the fifth pear category.”
4. L 45: This is not true: volatile compounds in pear have been studied even before the introduction of the spme.
Answer: We admired your scientific and rigorous attitudes. After we carefully read the reference [2], we learned that methods such as direct solvent extraction, distillation, simultaneous distillation-extraction (SDE) and supercritical fluid extraction (SFE) had been applied in isolation of volatiles from pears. Since 1994, SPME has been applied in fruit flavor studies. We have re-written this sentence to make the expression clear.
The sentence is “Volatiles emitted from pear fruits have always been studied by SPME and gas chromatography-mass spectrometry (GC-MS) in recent years.”
[2]. Rapparini, F; Predieris, S. Pear fruit volatiles. Hort. Rev. 2003, 28, 237-324.
5. L 78-79 & L 202-203: please report the length of the coating.
Answer: “The length of the fibre coating is 1 cm.” was added.
6. L 80-84: It is not clear which samples were analysed for comparison between fibres: the same sample with all the fibres or the number of volatile compounds (and the areas) are from the combination of fibres with all pear cultivars?
Answer: We are very sorry for our unclear expression. All SPME fibres used a same sample (yali pear, Pyrus bertschneideri Reld) for comparison the number and the areas of volatile compounds.
The sentence “The aroma components of a same yali pear (Pyrus bertschneideri Reld) were analyzed for fibres comparison.” was added in section 2.2.
L 84: “from yali pear” was added.
L 85: “the” was deleted.
7. L 84-86: was the DVB/CAR/PDMS fibre used to generate data in table 2? Please clarify.
Answer: The DVB/CAR/PDMS fibre was used to generate data in Table 2 (Table 1 after we rearranged table orders according to comment 15). So the sentence ‘Therefore, the 50/30 μm DVB/CAR/PDMS fibre was selected for aroma extraction in this study.’ was added in manuscript.
8. L 90: the statement should be smoothed out: the "volatile compounds were tentatively identified". The only criterium used for identification was the ms spectra matching with NIST database (L 232-233). The use of retention index is necessary for the identification of the compounds. This is still more critical for the identification of terpenes and terpenoids that present very similar (if not identical) fragmentation pattern.
Answer: Line 90, “tentatively” was added. We know your meaning that the use of retention index is necessary for the identification of the compounds. In our study, to distinguish some similar aromas, the standards were used. For n-alkanes, which have the same fragment ions (i.e. m/z 43, 57, 71, 85, 99) at 70 eV, the extraction process of the standard for each alkane was performed to obtain their retention time. Therefore, the retention times of alkanes are identified. For terpenoids, we compare the retention times of many repetitions, they are consistent. Your comments can give guidance to our following work.
9. L 93: "classification" stand for "comparison"?
Answer: "classification" in L 93 is not stand for "comparison". According to functional group differences, the aromas of pears are classified into 8 chemical classes, including esters, alkenes, alkanes, arenes, aldehydes, ketones, alcohols and others compounds. We have re-written this sentence to make the expression clear.
“The number of chemical classes of each pear is shown in Figure 3.”
10. L 94 & fig. 4: it is not clear if data refers to the number of compounds identified or the concentration.
Answer: The data in figure 4 refers to the concentration of compounds identified. We have re-written this sentence according to your comments.
The expression after correction: “Figure 4 shows that the percentage contents of volatile compounds in pears are of large differences. Esters are the dominant aromas in pears, followed by alcohols and aldehydes.”
11. Table 2 should report the matching factor with the NIST and the retention times recorded for each compound as well as retention index at least for the 1st dimension.
Answer: ① The aroma extraction process was repeated for three times to guarantee the accuracy of the results. So data in Table 1 (Table 1 after we rearranged table orders according to comment 15) are means±SD of three replications. For each aroma, each repetition produced a matching factor with the NIST. When the similarities of three replications are all above 800, the aroma is confirmed. Although the similarity of each repetition exceeds 800, small differences are existed. In table 1, each pear has a corresponding matching factor. Therefore, the matching factors with the NIST were not added.
② The retention times of volatile compounds are added in Table 1. The retention index contributes to qualitative analysis. For n-alkanes, which have the same fragment ions (i.e. m/z 43, 57, 71, 85, 99) at 70 eV, the extraction process of the standard for each alkane was performed to obtain their retention time. Therefore, the retention times of alkanes are identified. For terpenoids, we compare the retention times of many repetitions, they are consistent. Your comments can give guidance to our following work.
12. L 141: 6-methyl-5-heptene-2-one can also originate from farnesene that is present in higher amount in the peel compared to the flesh.
Answer: Thanks for your comments. Your comments are in accordance with Reference 25 and 26.
“It has been reported that 6-methyl-5-heptene-2-one was present in higher amount in the peel compared to the flesh and was a degradation product of lycopene [16] or α-farnesene [25, 26]” were added.
Reference 25 and 26 were added in manuscript.
[25] Mir, N.A.; Beaudry, R. Effect of superficial scald suppression by diphenylamine application on volatile evolution by stored cortland apple fruit. J. Agric. Food Chem. 1999, 47, 7−11.
[26] Hui, W.; Niu, J.P.; Xu, X.Y; Guan, J.P. Evidence supporting the involvement of MHO in the formation of superficial scald in ‘Dangshansuli’ pears. Postharvest Biol. Technol. 2016, 121, 43–50.
13. L 184-198: It is not clear why PCA was performed and which is the usefulness for the discussion of the results. Which information provides?
Answer: We have re-written this part according to your suggestions. In our previous work, there were no real conclusions on the results obtained except for a brief description of the results. We tried to draw an explicit conclusion like reference 3 and 4, but the numbers of the volatile compounds are so many that we can not distinguish correctly. Therefore, according to your comment and reference 16, the cluster analysis was performed.
[3] Chen, Y.Y.; Yin, H.; Wu, X.; Shi, X.J.; Qi, K.J.; Zhang, S.L. Comparative analysis of the volatile organic compounds in mature fruits of 12 Occidental pear (Pyrus communis L.) cultivars. Sci. Hortic. 2018, 240, 239-248.
[4] Qin, G.H.; Tao, S.T.; Cao, Y.F.; Wu, J.Y.; Zhang, H.P.; Huang, W.J.; Zhang, S.L. Evaluation of the volatile profile of 33 Pyrus ussuriensis cultivars by HS-SPME with GC-MS. Food Chem. 2012, 134, (4), 2367-2382.
[16] Gokbulut, I.; Karabulut, I. SPME-GC-MS detection of volatile compounds in apricot varieties. Food Chem. 2012, 132, (2), 1098-1102.
The cluster analysis and figure 5 were added in section 2.4 to clear expression the goal and the conclusion.
“The cluster analysis based on concentrations of identified volatile compounds was performed using the SPSS Statistical 19.0 software. The dendrogram (Figure 5) shows that two main groups are distinguished. Longyuanyangli, which has the maximum aroma numbers and the highest concentrations, is separated from other pear cultivars. Packham, D Jules, Clapp and Stark constitute the second group. They are 4 Occidental pears which are introduced from abroad. Figure 5 shows that the D Jules and Stark have the slight differences compared with other cultivars. Many factors affect the volatile compounds composition of the fruits. In this study, the volatile compositions of pears were found to be considerably different.”
Figure 5 Dendrogram obtained from cluster analysis based on the identified volatile compounds.
14. L 201: description of sample preparation is limited. Were the fruits pealed? Which section (L 209) of the fruit was diced? How and how long were the fruits stored before the measurements?
Answer: For each pear, the core was removed and the peel was reserved. The skin and flesh of each pear was cut into cubes (0.5 cm×0.5 cm×0.5 cm). The correction was presented in manuscript. All fruits were stored at 1 ℃ before experiments. For each cultivar in this study, after-ripening process was necessary to enhance the flavor and taste. Samples were placed at room temperature before experiments (approximately five days) to attain the state of consumption.
15. L 201-202: Table 1 is reported after table 2. Information reported is not complete. Were the fruits comparable in terms of ripening? Were the fruits harvested and stored in comparable conditions?
Answer: We are sorry that we mixed the sequential order of Tables. The order of the tables is rearranged in manuscript. The investigations with regard to ripening, harvest and storage conditions were carried out in other experiments. The proper time of harvesting and after-ripening process was important to guarantee the optimum quality of pears. In this study, the conventional indicators such as growing period, external morphology and skin color were used to judge the maturity of each cultivar. Each cultivar was picked by local growers of years experiences. All fruits were stored at 1 ℃ before experiments. For each cultivar in this study, after-ripening process was necessary to enhance the flavor and taste. Samples were placed at room temperature before experiments (approximately five days) to attain the state of consumption.
16. L 210-211: Was the reproducibility of the method tested? if so, please reports the results for the 2-nonanone.
Answer: We tested the reproducibility of the method. The RSD of the 2-nonanone was lower than 8.0% for each pear. The SD of each volatile compound was shown in Table 2.
17. L 213-214: The split mode is un-usual for SPME, there is a reason for this choice?
Answer: In comparison with one-dimensional gas chromatography (1D-GC), comprehensive two-dimensional gas chromatography (GC×GC) has higher sensitivities. The splitless mode was also researched in this study. But the peak area was very high so that the peak shape was not spindle-shaped and the combination of peaks was needed, which had a great influence on the quantitative analysis. To ensure the peak shape of the volatile compounds and the accuracy of the quantitative analysis, the split mode was used in this study. The split mode of 2:1, 5:1, 10:1, and 20:1 were studied in this study. Finally, the split mode of 10:1 was selected.
18. L 228-229: "acquired" is more appropriate than "scanned" when referring to a ToF MS. Please correct it.
Answer: The statements of “scanned” were corrected as “acquired”.
19. L 233: please provide NIST version.
Answer: Thanks to the Reviewer’s suggestion, the NIST 2017 was used in this study. We have added the “NIST2017” in L233 to explain the NIST version.
We admire your rigorous attitude and professional knowledge. It can give guidance to our following work. Once again, thank you very much for your comments and suggestions.

Reviewer 3 Report
This manuscript describes the SPME-GCxGC-TOFMS analysis of the aroma of pears. There are two major concerns that lead me to recommend a major revision.
My main concern is with the quantitative analysis. in the absence of actual standards, the qauntitation is an estimate only. There is no accounting for differences in detector response form compound to compound therefor the peak areas can only be used estimate the actual mass of each component. This must be made clear or the paper should be rewritten as qualitative analysis only with just some discussion of the relative amounts of major components.
A second concern is that usually in GCxGC papers, a chromatogram of the full analysis is provided, with important analytes or analyte classes indicated.
Author Response
Journal: Molecules
Manuscript ID: molecules-435985
Type: Article
Title: "Analysis of volatile compounds in pears by HS-SPME-GC×GC-TOFMS"
Correspondence Author: Zi-lei Chen ([email protected]) and Guo-sheng Yang ([email protected])
#Reviewer 3
1. My main concern is with the quantitative analysis. in the absence of actual standards, the qauntitation is an estimate only. There is no accounting for differences in detector response form compound to compound therefor the peak areas can only be used estimate the actual mass of each component. This must be made clear or the paper should be rewritten as qualitative analysis only with just some discussion of the relative amounts of major components.
Answer: Relative concentrations have been calculated referring to an internal standard of 2-nonanone. Therefore, in this manuscript, the concentrations all refer to relative concentration. According to references 3, 4, 15 and 16, the concentrations also refer to relative concentrations.
“The quantitative analysis for aroma components was carried out by internal standard method using 2-nonanone as an internal standard. Therefore, the concentration of each volatile compound was normalized to that of 2-nonanone.” in manuscript to explain relative quantification.
[3]. Chen, Y.Y.; Yin, H.; Wu, X.; Shi, X.J.; Qi, K.J.; Zhang, S.L. Comparative analysis of the volatile organic compounds in mature fruits of 12 Occidental pear (Pyrus communis L.) cultivars. Sci. Hortic. 2018, 240, 239-248.
[4]. Qin, G.H.; Tao, S.T.; Cao, Y.F.; Wu, J.Y.; Zhang, H.P.; Huang, W.J.; Zhang, S.L. Evaluation of the volatile profile of 33 Pyrus ussuriensis cultivars by HS-SPME with GC-MS. Food Chem. 2012, 134, 2367-2382
[15]. Yang, C.X.; Wang, Y.J.; Liang, Z.C.; Fan, P.G.; Wu, B.H.; Yang, L.; Wang, Y.N.; Li, S.H. Volatiles of grape berries evaluated at the germplasm level by headspace-SPME with GC-MS. Food Chem. 2009, 114, (3), 1106-1114.
[16]. Gokbulut, I.; Karabulut, I. SPME-GC-MS detection of volatile compounds in apricot varieties. Food Chem. 2012, 132, (2), 1098-1102.
2. A second concern is that usually in GCxGC papers, a chromatogram of the full analysis is provided, with important analytes or analyte classes indicated.
Answer: Thanks for your suggestion. Considering the resolution and size of the chromatogram, the chromatograms were put in supporting information.
“The 2D chromatography of five pears obtained after HS-SPME-GC×GC-TOFMS analysis is shown in Fig. S1-5. The colour gradient reflects the intensity of the TOFMS signal from low (blue) to high (red).” were added in section 2.3.
Fig. S1. 2D chromatogram (total ion chromatography) of Longyuanyangli.
Fig. S2. 2D chromatogram (total ion chromatography) of Packham.
Fig. S3. 2D chromatogram (total ion chromatography) of D Jules.
Fig. S4. 2D chromatogram (total ion chromatography) of Clapp.
Fig. S5. 2D chromatogram (total ion chromatography) of Stark.
Round 2
Reviewer 1 Report
The authors answered well to the comments made. Addition of the table (as supplementary material) given as a answer in question 2 could improve the reading and understanding.
Cluster analysis is interesting and adds more value than previous PCA analysis. I have just a comment : there is no label to the top axis of your dendogram. Moreover, if you want to help the reader to understand rapidly the diagram, I would suggest to change the colors. I may not be the only one to observe, at first sight, squares instead of lines.
Besides, in order to answer to some comments of the other reviewers, you may consider to publish your raw data.
Author Response
1. In order to improve reading and understanding, the table (given as a answer in question 2) was added in supplementary material.
Table S1 The peak numbers of different classes obtained using 4 SPME fibres.
50/30 μm DVB/CAR/PDMS | 85 μm PA | 65 μm PDMS/PVB | 100 μm PDMS | |
Esters | 67 | 49 | 62 | 46 |
Alkenes | 19 | 3 | 27 | 12 |
Alkanes | 19 | 4 | 17 | 14 |
Arenes | 20 | 2 | 15 | 0 |
Aldehydes | 9 | 6 | 10 | 2 |
Ketones | 5 | 2 | 4 | 4 |
Alcohols | 9 | 7 | 7 | 4 |
Others | 15 | 8 | 4 | 2 |
Total | 163 | 81 | 146 | 84 |
2. The dendrogram (Figure 5) was modified as follows:

Reviewer 2 Report
I appreciate the authors' tentative to improve the manuscript. The main drawback of the experimental design remains (answers 1, 2, 3 and 4 to main comments).
Author Response
Thanks for your suggestions.
We have carefully read reference 3 and 4. And we think that although many factors affect volatile compounds composition, the cultivar plays an important role. Investigations with regard to geographical locations and harvest and storage conditions were carried out and in other experiments.
In regard to comment 3 and 4, detail answers were presented in specific comments. We are sorry for our unclear explanation. Your comments can give guidance to our following work.
[3] Chen, Y.Y.; Yin, H.; Wu, X.; Shi, X.J.; Qi, K.J.; Zhang, S.L. Comparative analysis of the volatile organic compounds in mature fruits of 12 Occidental pear (Pyrus communis L.) cultivars. Sci. Hortic. 2018, 240, 239-248.
[4] Qin, G.H.; Tao, S.T.; Cao, Y.F.; Wu, J.Y.; Zhang, H.P.; Huang, W.J.; Zhang, S.L. Evaluation of the volatile profile of 33 Pyrus ussuriensis cultivars by HS-SPME with GC-MS. Food Chem. 2012, 134, (4), 2367-2382.
Reviewer 3 Report
The authors have adequately addressed the review comments.
Author Response
Thanks to your comments.